# Biaxial Structures of Localized Deformations and Line-like Distortions in Effectively 2D Nematic Films

**DOI:** 10.3390/nano14030246

**Published:** 2024-01-23

**Authors:** Luka Mesarec, Samo Kralj, Aleš Iglič

**Affiliations:** 1Laboratory of Physics, Faculty of Electrical Engineering, University of Ljubljana, 1000 Ljubljana, Slovenia; ales.iglic@fe.uni-lj.si; 2Department of Physics, Faculty of Natural Sciences and Mathematics, University of Maribor, 2000 Maribor, Slovenia; samo.kralj@um.si; 3Condensed Matter Physics Department, Jožef Stefan Institute, 1000 Ljubljana, Slovenia; 4Laboratory of Clinical Biophysics, Faculty of Medicine, University of Ljubljana, 1000 Ljubljana, Slovenia

**Keywords:** nematic shells, orientational order, topological defects, stomatocytes, order reconstruction mechanism

## Abstract

We numerically studied localized elastic distortions in curved, effectively two-dimensional nematic shells. We used a mesoscopic Landau-de Gennes-type approach, in which the orientational order is theoretically considered by introducing the appropriate tensor nematic order parameter, while the three-dimensional shell shape is described by the curvature tensor. We limited our theoretical consideration to axially symmetric shapes of nematic shells. It was shown that in the surface regions of stomatocyte-class nematic shell shapes with large enough magnitudes of extrinsic (deviatoric) curvature, the direction of the in-plane orientational ordering can be mutually perpendicular above and below the narrow neck region. We demonstrate that such line-like nematic distortion configurations may run along the parallels (i.e., along the circular lines of constant latitude) located in the narrow neck regions of stomatocyte-like nematic shells. It was shown that nematic distortions are enabled by the order reconstruction mechanism. We propose that the regions of nematic shells that are strongly elastically deformed, i.e., topological defects and line-like distortions, may attract appropriately surface-decorated nanoparticles (NPs), which could potentially be useful for the controlled assembly of NPs.

## 1. Introduction

Nature displays diverse soft materials [1,2] that may exhibit axial (i.e., nematic) orientational orders where the topology of the system is spherical. Typical examples of systems of spherical topology possessing in-plane order are biological membranes [3,4,5,6,7,8,9,10,11,12] and nematic liquid crystalline (LC) shells [13,14,15]. These structures are expected to play an important role in natural [3] and man-made [15] applications. Furthermore, they exhibit a rich pallet of different universal behaviors [16], particularly with respect to their topology and topological defects (TDs) [17], which we address in this paper.

The simplest class of such systems, where the largest number of investigations have been carried out, are two-dimensional (2D) closed manifolds [3,4,5,13,14,15] exhibiting axial nematic in-plane order. Such systems are typically represented by closed biological membranes (e.g., vesicles) or LC structures (e.g., colloids). The biological or artificial liposome membrane is very thin, and is considered to be a 2D structure; however, a closed membrane (a cell or a vesicle) is a 3D structure embedded in 3D space. Similarly, thin LC film is a 2D structure that forms a closed 3D shell embedded in 3D space. Consequently, 2D closed manifolds of our interest were exposed to a 3D curvature field. In our paper, we henceforth refer to such systems as *nematic shells*. These systems are relatively easily accessible, mathematically and experimentally (biological membranes and thin LC films covering colloids). Furthermore, different ordering mechanisms that can be systematically studied in 2D [2,13,14] are also manifested in higher dimensional systems [18,19,20,21], which are generally more difficult to analyze mathematically [21,22]. Consequently, one can often exploit 2D systems as a testbed of universal phenomena that are observed in diverse, physically distinct systems.

Nematic order is commonly presented by the uniaxial nematic director field $\vec{n}$ [1]. This unit field points along the local uniaxial direction, where the states $\pm \vec{n}$ are physically equivalent. Closed 2D manifolds with nematic order unavoidably exhibit TDs [15]. The latter refer to topologically stable localized distortions [17], to which one can assign a discrete topological charge, which is a conserved quantity. In 2D, it is given by the winding number *m*, which measures the total reorientation of $\vec{n}$ divided by 2π on encircling the center of a defect counterclockwise. A locally topologically stable TD is characterized by a discrete value of the winding number [1]: $m\in \left\{\pm \frac{1}{2},\pm 1,\pm \frac{3}{2}\ldots \right\}.$ Note that half of the integers are possible due to the head-to-tail axial symmetry. Furthermore, defects bearing opposite charge signs tend to annihilate into a defectless state. One commonly refers to TDs bearing m>0 and m<0 as *defects* and *antidefects*, respectively. Common TD structures and their combinations are shown in Figure 1. The total winding number $m_{tot}$ of a closed 2D manifold is given by the Gauss–Bonnet and Poincare–Hopf theorems [23]: $m_{tot}=\frac{1}{2\pi }∯K_{g}d^{2}\vec{r}=2\left(1-g\right).$ Here, $K_{g}$ stands for the Gaussian curvature of a local surface patch of area $d^{2}\vec{r}$, whose surface integral is a conserved quantity, defined by the genus g (i.e., the number of holes) of a closed manifold surface. In the case of spherical geometry, it holds that g=0 and $m_{tot}=2.$ In common conditions, a locally enforced TD bearing $\left|m\right|>1/2$ tends to decay into its elementary units [24,25] bearing $m=m_{0},$ where $m_{0}=\pm 1/2,$ due to energy reasons. In addition, surface patches exhibiting $K_{g}>0$ ($K_{g}<0)$ attract [26] TDs bearing m>0 (m<0) due to the effective topological charge cancellation (ETCC) mechanism [27]. The latter determines the local impact of the so-called intrinsic curvature [23], which is independent of how a 2D manifold is embedded in 3D Euclidean space. The effect of the 3D curvature field is measured by the so-called extrinsic curvature [3,4,5,6,10,28,29,30], also referred to as the deviatoric curvature [4,5,6,28,31,32,33,34,35,36]. In its simplest form, the extrinsic curvature is quantified by the difference between the local principal curvatures, and in general tends to align the local nematic ordering along the direction exhibiting the lower curvature. However, if one also considers the intrinsic curvature of the molecules exhibiting nematic order (e.g., curved proteins or nematics), the preferred average orientation of molecules can be at any angle. In such cases, molecules tend to orient in the direction at which they best fit onto the curved surface, which was studied in [10] by taking into account the deviatoric bending energy of the surface-attached curved nematic molecules, originally introduced in [36].

In effectively 2D systems (i.e., the relevant variational parameters exhibit essential spatial variations only along two coordinates of a 3D system), TDs commonly exhibit point-like structures [14,15,26,29]. Analysis of TDs in 3D reveals that the core structure of such defects, whose local structure is described by the local winding number [1,17] m=±1/2, is strongly biaxial [18], where the defect core center exhibits negative uniaxial order. To describe such configurations mathematically, one needs to introduce the nematic tensor order parameter $\underline{Q}$ [1]. On crossing the defect center, $\underline{Q}$ exhibits the exchange of its eigenvalues [18]. Later, the described mechanism, which mediates relatively large orientational conflicts imposed on a distance comparable to the biaxial order parameter correlation length $\xi_{b}$, was dubbed the order reconstruction (OR) mechanism [19,20,21]. This mechanism was originally introduced to explain line-like distortions in thin hybrid nematic cells [19] of thickness $h\sim \xi_{b}.$

In this research, we show that OR-type distortions can enable localized line-like distortions bearing m=0 in 2D nematic shells strongly curved in 3D space.

## 2. Phenomenological Model of Nematic Ordering

### 2.1. Order Parameter

Within the mesoscopic Landau-de Gennes-type approach, nematic orientational order is commonly presented by tensor nematic order parameters. In 3D, it is generally expressed in terms of five variational parameters. In 2D systems, nematic order is commonly presented only in terms of two variational parameters. In our contribution we show that in general, key features of TDs in thin LC films are nevertheless well described by such 2D approaches.

In three-dimensional (3D) space, nematic order within the Landau-de Gennes-type approach is represented by the following traceless and symmetric tensor nematic order parameter [22]:(1)$$\underline{Q}=\sum_{i=1}^{3}s_{i}{\vec{e}}_{i}⊗{\vec{e}}_{i},$$
where $s_{i}$ and ${\vec{e}}_{i}$ stand for the $\underline{Q}$ eigenvalues and eigenvectors, respectively. Uniaxial states are commonly expressed as follows [1]:(2)$$\underline{Q}=s\left(\vec{n}⊗\vec{n}-\underline{I}/3\right)$$
where this is expressed in terms of the nematic director field $\vec{n}$ and the nematic uniaxial order parameter $s=\in \left\{-\frac{1}{2},1\right\}.$ Here, s>0 and s<0 geometrically mimic prolate- and oblate-like mesoscopic orders, respectively (these structures are characterized by different fluctuation probability distributions [1]), which is absent for s=0. In bulk, equilibrium nematic order is commonly positively uniaxial. In case of elastic distortions, $\underline{Q}$ can enter biaxial states [24]. The degree of biaxiality is measured by the following biaxial parameter [37,38]:(3)$$\beta^{2}=1-\frac{6\;{\left(Tr{\underline{Q}}^{3}\right)}^{2}}{{\left(Tr{\underline{Q}}^{2}\right)}^{3}}\in \left[0,1\right].$$
where $\beta^{2}=0$ and $\beta^{2}=1$ reflect states exhibiting uniaxial states and maximal biaxiality, respectively [37].

Note that in general, due to the condition $Tr\underline{Q}=s_{1}+s_{2}+s_{3}=0$, only two “amplitude” parameters are needed to describe the eigenvalues of $\underline{Q}$. A convenient parametrization of eigenvalues is expressed by the biaxiality angle γ and the effective amplitude $s_{0}$, as follows [39]:(4a)$$s_{1}=\frac{2s_{0}}{3}\cos\gamma ,\;s_{2}=-\frac{2s_{0}}{3}\cos\left(\gamma -\frac{\pi }{3}\right),\;s_{3}=-\frac{2s_{0}}{3}\cos\left(\gamma +\frac{\pi }{3}\right),$$
(4b)$$s_{0}=\sqrt{\frac{3}{2}Tr{\underline{Q}}^{2}}.$$

For a fixed nematic order parameter tensor $\underline{Q}$ eigen-frame {${\vec{e}}_{1},{\vec{e}}_{2},{\vec{e}}_{3}$}, the possible nematic configurations on varying $s_{0}$ and γ are depicted in Figure 2a. The melted isotropic state (in which orientational order is absent) refers to the point $s_{0}=0$. Configurations determined by $\gamma =0,\gamma =-\frac{2\pi }{3}$ and $\gamma =\frac{2\pi }{3}$ (γ=π,γ=π/3 and γ=−π/3) correspond to positive (negative) uniaxial states aligned along ${\vec{e}}_{1},{\vec{e}}_{2}$ and ${\vec{e}}_{3},$ respectively. The degree of biaxiality attains its maximum for γ=±π/6, γ=±π/2 and γ=±5π/6.

In the following, we consider cases where the nematic order is confined in the 2D plane {${\vec{e}}_{1},{\vec{e}}_{2}$} of the 3D space. In such cases, one $\underline{Q}$ eigenvalue always points along ${\vec{e}}_{3}$, which coincides with the local surface normal of a 2D submanifold. A convenient representation in terms of variational parameters {$q_{1}{,q}_{2},q_{3}$} reads as follows [39]:(5)$$\underline{Q}=\left(q_{3}+q_{1}\right)\left({\vec{e}}_{1}⊗{\vec{e}}_{1}\right)+\left(q_{3}-q_{1}\right)\left({\vec{e}}_{2}⊗{\vec{e}}_{2}\right)+q_{2}\left(\left({\vec{e}}_{1}⊗{\vec{e}}_{2}\right)+\left({\vec{e}}_{2}⊗{\vec{e}}_{1}\right)\right)-{2q}_{3}\left({\vec{e}}_{3}⊗{\vec{e}}_{3}\right)={\underline{Q}}_{u}^{\left(3D\right)}+\underline{q}.$$
where ${\underline{Q}}_{u}^{\left(3D\right)}$ stands for a 3D uniaxial order parameter, and $\underline{q}$ for an effectively 2D order parameter, as follows:(6a)$${\underline{Q}}_{u}^{\left(3D\right)}=q_{3}\left(\left({\vec{e}}_{1}⊗{\vec{e}}_{1}\right)+\left({\vec{e}}_{2}⊗{\vec{e}}_{2}\right)\right)-{2q}_{3}\left({\vec{e}}_{3}⊗{\vec{e}}_{3}\right),$$
(6b)$$\underline{q}=q_{1}\left(\left({\vec{e}}_{1}⊗{\vec{e}}_{1}\right)-\left({\vec{e}}_{2}⊗{\vec{e}}_{2}\right)\right)+q_{2}\left(\left({\vec{e}}_{1}⊗{\vec{e}}_{2}\right)+\left({\vec{e}}_{2}⊗{\vec{e}}_{1}\right)\right).$$

Note that ${\underline{Q}}_{u}^{\left(3D\right)}$ exhibits negative and positive uniaxiality for $q_{3}>0$ and $q_{3}<0$, respectively (for $q_{3}=0$, the system exhibits maximal biaxiality). In terms of parameters {$q_{1},q_{2},q_{3}$}, the degree of biaxiality reads as follows:(7)$$\beta^{2}=\frac{(q_{1}^{2}+q_{2}^{2}){\left(q_{1}^{2}+q_{1}^{2}-9q_{3}^{2}\right)}^{2}}{{\left(q_{1}^{2}+q_{1}^{2}+3q_{3}^{2}\right)}^{3}}.$$

In this parametrization, $\underline{Q}$ eigenvalues are expressed as follows:(8a)$$s_{1}=q_{3}+\sqrt{q_{1}^{2}+q_{2}^{2}},$$
(8b)$$s_{2}=q_{3}-\sqrt{q_{1}^{2}+q_{2}^{2}},$$
(8c)$$s_{3}=-2q_{3}.$$

Note that the exchange of eigenvalues $s_{1}↔s_{2}$ coincides with the condition $\sqrt{q_{1}^{2}+q_{2}^{2}}=0.$ Furthermore, configurations with $q_{3}=0$ display maximal biaxiality (i.e., $\beta^{2}=1$).

### 2.2. Order Reconstruction

The OR mechanism is activated when a large enough orientational frustration is imposed on a distance that is comparable to the biaxial coherence length $\xi_{b}$ [21,40]. The latter measures a characteristic distance on which locally imposed biaxial order decays to essentially uniaxial order in bulk uniaxial LC shells. Within this mechanism, a relatively localized LC transformation is achieved by changes in order parameter amplitude space, while the $\underline{Q}$ framework can remain fixed.

A representative transformation is shown in Figure 2. In Figure 2a, we present a system subjected to conflicting boundary condition. It illustrates a uniaxial orientational variation within an LC body confined in a plane-parallel cell along the z-axis of the (x,y,z) Cartesian system, where the confining planes at z=0 and z=h strongly impose $\vec{n}\left(z=0\right)={\vec{e}}_{x}$ and $\vec{n}\left(z=h\right)={\vec{e}}_{z}$. At the lateral boundaries, one assumes free boundary conditions. If h> $\xi_{b}$, the imposed frustration can be resolved by gradual reorientation of $\vec{n}$, whereby the nematic states everywhere exhibit essentially uniaxial order. These changes correspond to the rotation of the amplitude parameter space frame in Figure 2b, where the state at z=0 corresponds to $\vec{n}\left(z=0\right)={\vec{e}}_{x}={\vec{e}}_{1}$, and on increasing z, the eigenvector ${\vec{e}}_{1}$ gradually reorients into $\vec{n}\left(z=0\right)={\vec{e}}_{z}.$ On the other hand, the imposed frustration could be realized via the OR mechanism [19] when ${h\sim \xi }_{b}$, which is illustrated in Figure 2b–d. In these figures, we label representative states of the transformation with the numbers 1 to 5. During the transformation, the amplitude order parameter remains fixed, i.e., ${\vec{e}}_{x}={\vec{e}}_{1},$ ${\vec{e}}_{y}={\vec{e}}_{2}$ and ${\vec{e}}_{z}={\vec{e}}_{3}$, which is valid for $z\in \left[0,h\right]$. Consequently, biaxial states must be realized within the distance *h*, which is shown in Figure 2d. In Figure 2c, we plot the corresponding geometric presentation of mesoscopic states. Note that OR does not require melting of nematic order. In the case shown in Figure 2b–d, we set that $s_{0}$ remains constant during the transformation. This is justifiable, deep in the nematic phase [38,41]. One sees that due to topological reasons, the negative uniaxial state along ${\vec{e}}_{y}$ must be realized in between (number 3 in Figure 2b–d). Furthermore, this state takes place between the two regions exhibiting maximal biaxiality (i.e., $\beta^{2}=1,$ see the numbers 2 and 4 in Figure 2b–d). Note that uniaxial states exhibit locally cylindrical symmetry, which is broken in biaxial states.

### 2.3. 2D Model of Nematic Ordering

In 2D, we describe the nematic order by the effectively 2D tensor order parameter field $\underline{q}$ (Equation (6b)). Therefore, we neglect spatial variations in $q_{3}$; see Equations (5) and (6). In general, this is justifiable. Namely, studies [38,39,42] in 3D systems that exhibit similar orientation frustrations as we consider in the present study, reveal that $q_{3}$ generally decreases on crossing a defect core. However, these variations do not qualitatively affect the phenomena of our interest.

In its eigen-frame, determined by the nematic director field $\vec{n}$, 2D tensor order parameter field $\underline{q}$ can be expressed within the Landau-de Gennes-type approach as follows [43]:(9)$$\underline{q}=s\left(\vec{n}⊗\vec{\;n}-{\vec{n}}_{⊥}⊗{\vec{n}}_{⊥}\right),$$
where $\{\vec{n},$ ${\vec{n}}_{⊥}\}$ are its eigenvectors with the corresponding eigenvalues {s, −s}. Here, s∈[0, 1/2] is the amplitude of nematic order in an infinitesimally small part of the surface determined by the surface normal $\vec{v}=\vec{n}\times {\vec{n}}_{⊥}.$ In terms of tour parametrization (see Equation (5)), the following holds:(10)$$s=\sqrt{q_{1}^{2}+q_{2}^{2}}.$$

Therefore, the order reconstruction in 3D space is realized when s=0 in the 2D submanifold. Furthermore, studies on similar systems [39] reveal that $q_{3}$ exhibits a relatively weak quantitative (it does not change its sign) change in amplitude in the eigenvalues $s_{1}↔s_{2}$ exchange. For this reason, we assume a constant value of $q_{3}$ and numerically determine the remaining variational parameters {$q_{1},q_{2}$} using a 2D model in the corresponding 2D submanifold. Then, the relevant order parameter is given by Equation (6b). To visualize our results in a common 3D presentation given by Equation (5), we impose a finite value of $q_{3}=0.04$, which is suggested by our previous simulations for similar conditions [39]. Note that $q_{3}=0$ would correspond to a state exhibiting maximal biaxiality (see Equation (7)).

The free-energy density $f=f_{c}+f_{e}$ consists of the condensation ($f_{c}$) and elastic ($f_{e}$) contributions, as follows [3]:(11a)$$f_{c}=a_{0}\left(T-T^{*}\right)\;Tr{\;\underline{q}}^{2}+\frac{b}{4}{\left(Tr\;{\underline{q}}^{2}\right)}^{2},$$
(11b)$$f_{e}=\frac{1}{2}k_{i}Tr{\left({\vec{\nabla }}_{s}\;\underline{q}\right)}^{2}+k_{e}Tr(\underline{q}{\underline{C}}^{2}).$$

In Equation (11a), the quantities $a_{0},\;b$ and $T^{*}$ are positive phenomenological constants. The condensation term (Equation (11a)) enforces nematic orientational order below the critical temperature $T_{c}$ (in a flat geometry $T_{c}=T^{*}$). The degree of equilibrium nematic order for ${T<T}^{*}$ is given by $s_{0}=\sqrt{a_{0}(T^{*}-T)/b}$. The elastic free energy term (Equation (11b)) consists of the {intrinsic, extrinsic} curvature contributions weighted by positive elastic moduli {$k_{i},\;k_{e}$}. In Equation (11b), $\underline{C}={\;\sigma }_{1}{\vec{e}}_{1}⊗{\vec{e}}_{1}+{\;\sigma }_{2}{\vec{e}}_{2}⊗{\vec{e}}_{2}$ stands for the curvature tensor, where $\sigma_{1}$ $\left(\sigma_{2}\right)$ is the principal curvature along ${\vec{e}}_{1}$ (${\vec{e}}_{2}$) [3].

Equilibrium nematic textures are calculated on fixed closed 2D manifolds, where $T<T_{c}$. We consider axially symmetric shapes of spherical geometry with the surface area A. In a 3D Cartesian system (x,y,z), defined by the unit vectors {${\vec{e}}_{x},{\vec{e}}_{y},{\vec{e}}_{z}\}$, the position vector $\vec{r}$, defining such shapes, is given by the following [3,10,44]:(12)$$\vec{r}=\rho \left(l\right)\cos\left(\varphi \right){\vec{e}}_{x}+\rho \left(l\right)\sin\left(\varphi \right){\vec{e}}_{y}+z\left(l\right){\vec{e}}_{z},$$
where $\rho \left(l\right)$ and z(l) are the shape profile coordinates in the (ρ,z)-plane. Here, φ∈[0, 2π] is the azimuthal angle and l is the arc length of the profile curve. The total length of the profile curve is denoted by $L_{s}$. All of the distances in our model are scaled with respect to $R=\sqrt{A/(4\pi )}$ [10], which is the radius of a sphere with the same surface area A as the surface area of the investigated shape. The tensor order parameter s is scaled with respect to the bulk equilibrium order parameter $s_{0}$. The typical distance, on which a locally perturbed nematic amplitude is recovered, is estimated by the following nematic order parameter correlation length [3]:(13)$$\xi =\sqrt{k_{i}/(a_{0}(T^{*}-T))}.$$

## 3. Results and Discussion

In Figure 3, we plot a reference configuration on a spherical shape in terms of 2D- (Figure 3a,b) and 3D (Figure 3c,d)-order parameters. The equilibrium configuration hosts four m=1/2 TDs that reside in the vertices of a hypothetical inscribed tetrahedron (Figure 3a,b) to maximize their separation, since TDs with the same charge repel each other [26,27]. The extrinsic curvature contribution (weighted by $k_{e}$ in Equation (11b)) does not have any effect on isotropic surfaces. Therefore, on a spherical surface presented in Figure 3, the configuration is the same for any value of $k_{e}$. In the center of TDs, the 2D amplitude of nematic order is melted (i.e., s=0, see red patches in Figure 3a,b). In 3D presentation, these points correspond to negative uniaxial order parameters (see Figure 3d). The degree of biaxiality $\beta^{2}$ (Figure 3c,d) is low on most of the surface because the condensation free-energy term favors uniaxial states. The cores of TDs exhibit uniaxial states, as evidently shown in Figure 3d. The centers of defects are surrounded by strongly biaxial states exhibiting volcano-like profiles, with circular rings exhibiting maximal biaxiality in $\beta^{2}$ presentation.

In Figure 4, we present the orientational ordering configuration and the degree of biaxiality on a stomatocyte (invaginated) 3D shape of a closed, thin 2D nematic shell. The orientational ordering on this shape is highly affected by the extrinsic (deviatoric) curvature contribution weighted by $k_{e}$ in Equation (11b), because the shape contains many anisotropic surface curvature regions where the principal curvatures differ, i.e., ${\;\sigma }_{1}\neq {\;\sigma }_{2}$. The extrinsic term is minimized when the molecules are locally oriented along the principal curvature direction with the lower absolute value of the principal curvature. This tendency is clearly demonstrated in Figure 4a,b, where different orientations are enforced below and above the stomatocyte neck. Below the neck (on the stomatocyte outer surface), molecules are oriented perpendicular to the shape’s axis, as that is the direction of lower principal curvature. On the other hand, above the neck (on the invagination), molecules are oriented parallel to the shape’s axis because the shape is less curved in that direction. Consequently, we observe a rapid change in the orientation of molecules in the neck. Such a rapid change requires local melting of orientational order (s=0, see the red line in Figure 4a,b), giving rise to a localized line-like distortion without the topological charge (m=0). Additionally, we have two m=1/2 TDs on the outer stomatocyte surface, and two m=1/2 TDs on the invagination part of the surface, so that the total topological charge $m_{tot}=2$ is dictated by topology (Gauss–Bonnet and Poincare–Hopf theorems [23]). Note that the above-described orientation preference of molecules above and below the stomatocyte (cup shape) neck region and the consequent formation of a localized line-like distortion in the neck are valid if the outer part of the stomatocyte surface is qualitatively different from the invagination part of the surface. In our case, the outer part of the stomatocyte is slightly oblate, while the invagination is slightly prolate (see Figure 4b). Furthermore, these orientation preferences are true for non-curved molecules described by the extrinsic term given in Equation (11b). If the molecules were curved, their orientation preference would also depend on their intrinsic curvature, as predicted in [10], by considering the deviatoric bending energy of curved nematic molecules attached to the surface [36].

Again, the degree of biaxiality around the TDs is at the maximum, as previously described and discussed in Figure 3 (yellow rings surrounding the TDs in Figure 4d). As for the line-like distortion in the neck, we observe a high degree of biaxiality just above and below the neck (see Figure 4c,d). This wall-like biaxial profile is characteristic for the OR-type transformation. One sees that the line displaying uniaxial order lies between the two lines exhibiting maximal biaxiality.

### Coupling between Topological Defect and Nanoparticles

Next, we considered the impact of nanoparticles (NPs) on nematic ordering within nematic shells. Their influence depends on the dimensionless quantity $\mu =R_{NP}W/K,$ where $R_{NP}$ stands for the characteristic NP’s linear size, W measures the anchoring strength of the NP–LC interface, and K is the representative Frank elastic constant [1]. In the regimes μ≪1 and μ≫1, nanoparticles have negligible and strong effects [17] on the surrounding nematic order, respectively. NPs usually exhibit spherical topology (i.e., they do not have holes), and in the strong anchoring limit (i.e., μ≫1) they effectively act as topological point defects bearing topological charge one (therefore, m=1 in effectively 2D systems). The coupling of NPs with TDs is qualitatively different in the respective regimes. In the regime μ≪1, NPs are attracted to the cores of defects owing to the universal defect core replacement (DCR) mechanism [45,46,47,48]. Namely, the key local penalty of introducing a TD into a uniform LC order arises due to the energetically expensive, essentially melted defect core. In effectively 2D systems, this penalty is roughly given by $\Delta F\sim \left|f_{c}\right|V_{c},$ where the core volume is given by $V_{c}\sim \pi \xi^{2}d$, where d estimates the LC film thickness. If an NP is trapped within the defect core, and it negligibly influences $\hat{n}$ (i.e., it does not introduce additional LC elastic penalties), the core free-energy contribution reads $\Delta F\sim \left|f_{c}\right|{(V}_{c}-V_{NP}),$ where $V_{NP}$ is the NP’s volume and one assumes $R_{NP}\leq \xi$. Therefore, the core penalty is reduced because the melted regime is partially replaced by the NP’s volume. On the other hand, NPs characterized by μ≫1 tend to create additional TDs in LC media. For example, an NP exhibiting spherical topology would enforce either one m=−1 defect or two m=−1/2 defects in an initially uniform LC pattern within a flat LC film. In such a way, the total topological charge of the film would equal to zero, allowing the realization of effectively spatially homogeneous far-nematic director fields.

In Figure 5, we illustrate the impact of a 2D circular NP on an effectively 2D nematic order within a spherical nematic shell enforcing $m_{tot}=2$, where the NP–LC interaction is characterized by μ≪1 (Figure 5a,c) and μ≫1 (Figure 5b,d). Furthermore, we assume that the NP–LC interface favors homeotropic anchoring, and consequently the NP effectively acts as an m=1 point defect in the regime μ≫1. Note that $m_{tot}=m_{tot}^{\left(LC\right)}+m_{tot}^{\left(NP\right)}$ counts the total topological charge in the film, where $m_{tot}^{\left(LC\right)}$ and $m_{tot}^{\left(NP\right)}$ represent the total charge of “real” TDs realized in the LC body and the effective charge imposed by NPs, respectively. Individual “real” 2D TDs are commonly carriers of $\left|m\right|=1/2$ charges [24,25]. On the other hand, a circular-shaped NP is an m=1 carrier in the μ≫1 regime. The nematic pattern shown in Figure 5a (μ≪1) is essentially the same as in the absence of an NP (see Figure 3), although the total free energy of the system is slightly lowered (owing to the DCR effect). On the contrary, the pattern in Figure 5b (μ≫1) is qualitatively different. Such a configuration was also predicted in [27]. Namely, topology requests $m_{tot}=2$, and the system tends to exhibit minimal free energy. The resulting compromise is realized by introducing two m=1/2 TDs into the LC medium because $m_{tot}^{\left(NP\right)}=1$ (Figure 5b,d).

## 4. Conclusions

In the present study, we focused on order reconstruction-resolved frustrations in closed deformable nematic shells curved in 3D space. Sphere-like shells typically possess four m=1/2 point defects [27,43]. Strong local elastic distortions of orientational order within the cores of these defects can be realized by the OR mechanism. In such a way, large differences in orientational order are reflected on the biaxial order parameter length scale (typical of the order of 50 nm), which roughly determines the core size of point defects [49]. The center of each defect exhibits negative uniaxiality [18,50]. The local biaxial configuration of TDs has a volcano-like structure (see Figure 3 and Figure 4) characterized by circular rims of maximal biaxiality [38]. In addition, strongly deformed shells can exhibit line-like distortions in the neck regions exhibiting high curvature deviator values (Figure 4), where the order parameter exhibits an exchange of eigenvalues (i.e., OR transformation) on crossing them. We illustrated such structures in stomatocyte-type 3D shell shapes (configurations) with invaginations (Figure 4). This OR-type deformation displays a valley-like biaxial profile of $\beta^{2}$ spatial dependence, where the central line exhibits negative uniaxiality and is enclosed by two lines exhibiting maximal biaxiality (Figure 4c,d). Contrary to point defects, which bear topological charges $\left|m\right|$ = 1/2, the predicted narrow region of practically discontinuous change in the orientational ordering along the parallels (i.e., distortions in the narrow region at the border line between the two domains in orientational ordering) in the stomatocyte neck (Figure 4a,b) is without topological charge (chargeless). Note that this phenomenon of sharp discontinuous line changes in orientational/nematic ordering is mainly due to the head-to-tail nematic symmetry of molecules [51]. The present study suggests that one can realize point or line-like distortions in effectively 2D systems deformed in 3D space exhibiting nematic order.

Regarding the possible applications of the theoretical results presented in this paper, it should be pointed out that the predicted line-like distortion regions (distortion sites) could act as attractors for certain types of nanoparticles (NPs), exploiting the defect core replacement (DCR) mechanism [45,46,47]. According to the DCR mechanism, the condensation free-energy penalty (due to essentially melted LC structures or energetically unfavorable biaxial orders) can be reduced by replacing a part of a deformed LC with nanoparticles (NPs).

Strongly anisotropic neck regions (like the stomatocyte neck region in Figure 4) could be formed by encapsulation of spherical particles by the membrane [52], as schematically presented in Figure 6. In biological cells, endocytosis and phagocytosis represent similar processes. Wrapping of the membrane around particles [53,54,55,56,57,58,59] occurs when it is energetically favorable for the membrane to bind to the particle [60,61,62,63,64,65,66], i.e., when the binding/adhesion energy between the particle and the membrane is negative [67]. Even though the line-like distortion in the neck region of our stomatocyte presented in Figure 4 is not a TD, it represents a source of large local elastic penalties due to an essentially melted LC structure, which means that the local interactions between the neighboring nematic molecules within the neck are weakened. Consequently, the neck may rupture, leading to the membrane fission (vesiculation) process [10,68] where two distinct closed shells (i.e., parent and daughter shells) would be formed, as schematically presented in Figure 6e. The process of intercalating nanoparticles (for example small proteins) in the line-like distortion region in the membrane neck could further facilitate neck constriction, and consequently fission/vesiculation [69].

The intercalation of NPs in the region of TDs or line-like distortions could be very effective if the NP’s interface does not strongly distort the surrounding nematic director field [48] (i.e., the principal $\underline{Q}$ eigenvalue). In experiments, this could be enabled either using small enough NPs, or NPs coated with flexible polymer chains (the latter can rearrange to accommodate the surrounding LC orientation structure). Therefore, by manipulating TDs on nematic shells, one could control the positions of trapped NPs within them, which may be useful for diverse applications. If, for example, conductive NPs assemble within a localized line-like distortion [70], they could serve as a conducting wire. By controlling the position of line-like distortions, one could develop tunable metamaterials and rewirable nano- or micro-wires [71]. Furthermore, recent results reveal that line-like distortions could be rewired [71] using external electric fields. In these scenarios, controlled and pre-established arrays of line-like distortions would resemble a complex network of conductive nano- or micro-wires with rewiring capability, where various arrangements would correspond to different functionalities [71]. This may lead to a variety of applications in soft matrices such as multistable optical displays, electronics and charge carrier pathways for photovoltaics [71]. For experimental purposes, one could also use quantum dots as nanoparticles, as they can emit in the visible electromagnetic spectrum, making it easier to determine their location.

Furthermore, in biological systems, interactions between TDs and nanosized objects could play a vital role in several biological mechanisms [72,73]. For example, TDs in an epithelial cell organization are exploited to efficiently remove dead cells [73]. Therefore, they act as an attractor for qualitatively different cells with respect to the “healthy” background, which is certainly governed by the energy minimization process. Interactions between NPs and TDs may also play a role in cross-membrane transport. Namely, TDs introduce inhomogeneities, which could be exploited to facilitate the transport of nano-objects needed for the activation of certain biological cell processes [74,75].

## Figures and Tables

**Figure 1 nanomaterials-14-00246-f001:**
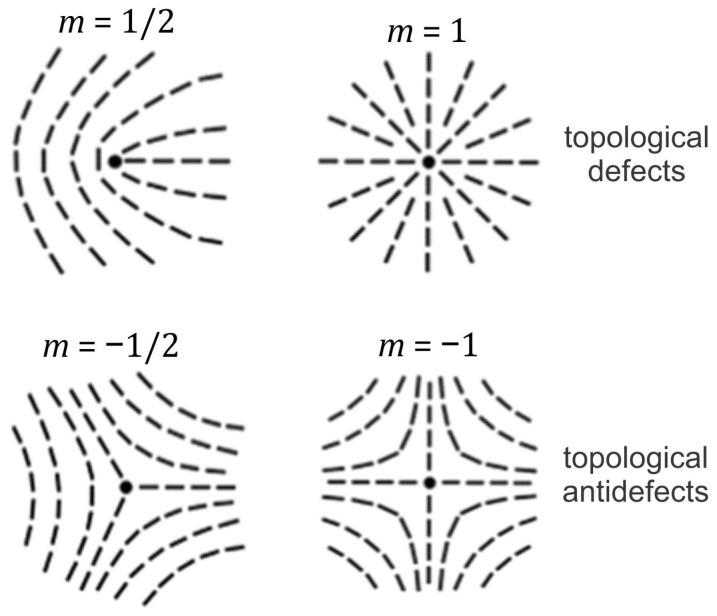
Typical topological defects and their charges in two dimensions.

**Figure 2 nanomaterials-14-00246-f002:**
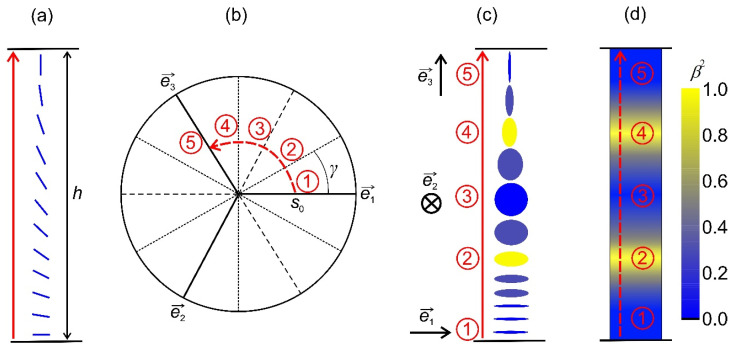
Schematic presentation of the order reconstruction (OR) mechanism. (**a**) System subjected to conflicting boundary condition at z=0 and z=h. The imposed frustration could be realized via the OR mechanism (**b**–**d**), where representative states of the transformation are labeled with the numbers 1 to 5. Due to topological reasons, the negative uniaxial state along ${\vec{e}}_{2}$ must be realized in between (**b**–**d**). Furthermore, this state takes place between the two regions exhibiting maximal biaxiality ($\beta^{2}=1$) (**c**,**d**).

**Figure 3 nanomaterials-14-00246-f003:**
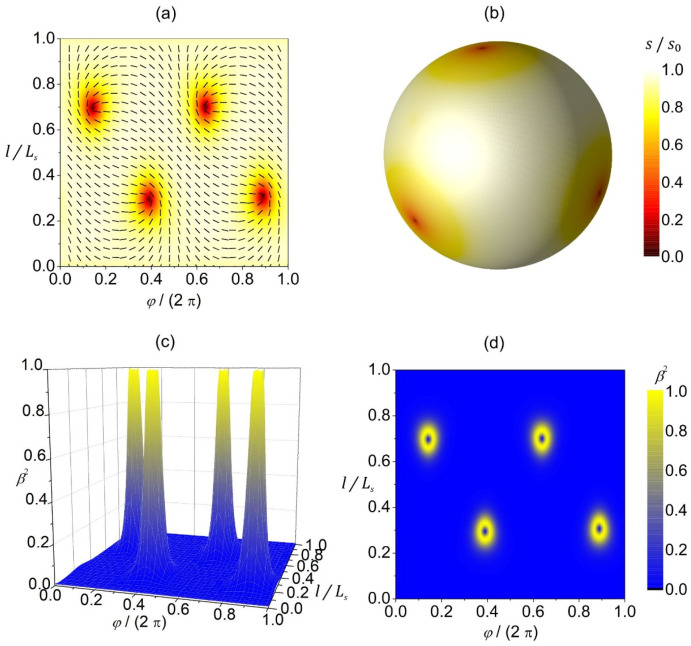
Orientational ordering configuration and the degree of biaxiality on a spherical shell. (**a**) The amplitude of nematic order $s/s_{0}$ (presented with color coding) and the orientation of molecules (denoted by the rods) in the (φ,l)-plane. (**b**) 3D visualization of the equilibrium texture of the amplitude of nematic order $s/s_{0}$. The degree of biaxiality $\beta^{2}$ presented as a 3D plot (**c**) and with color coding (**d**). $L_{s}$ stands for the length of the shape profile curve. The parameters used in the simulations are R/ξ=5, $k_{e}=k_{i}/2$, $q_{3}=0.04$.

**Figure 4 nanomaterials-14-00246-f004:**
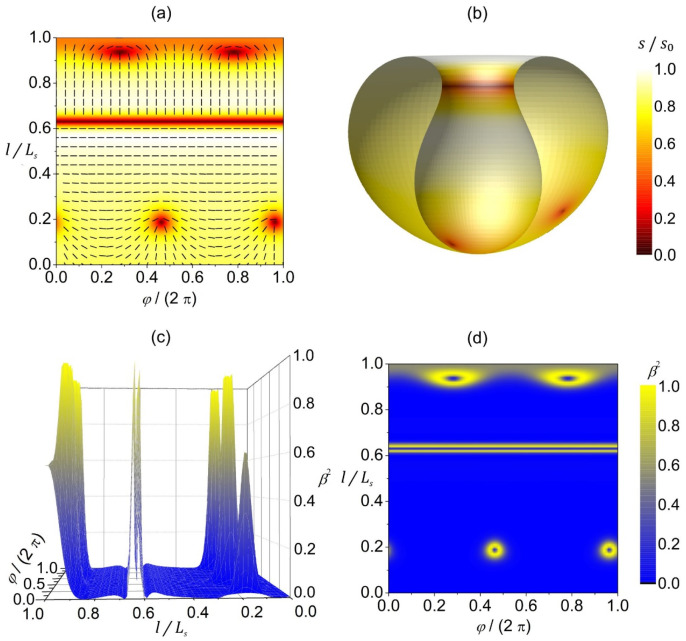
Orientational ordering configuration and the degree of biaxiality on a stomatocyte vesicle shape. (**a**) The amplitude of nematic order $s/s_{0}$ (presented with color coding) and the orientation of molecules (denoted by the rods) in the (φ,l)-plane. (**b**) Equilibrium texture of the amplitude of nematic order $s/s_{0}$ plotted on half of the stomatocyte shape. The degree of biaxiality $\beta^{2}$ presented as a 3D plot (**c**) and with color coding (**d**). $L_{s}$ stands for the length of the shape profile curve. The parameters used in the simulations are R/ξ=7, $k_{e}=k_{i}/2$, $q_{3}=0.04$.

**Figure 5 nanomaterials-14-00246-f005:**
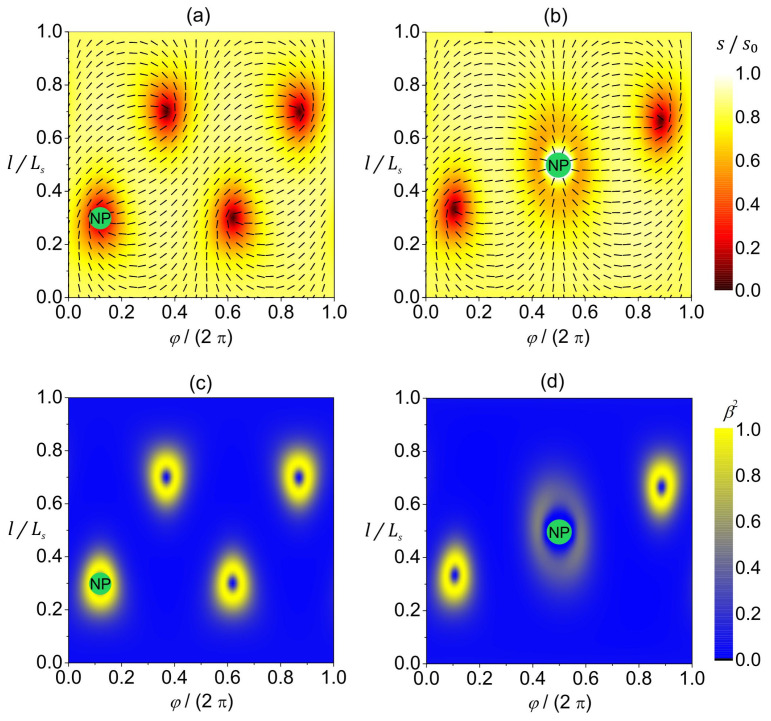
Orientational ordering configuration (**a**,**b**) and the degree of biaxiality (**c**,**d**) on a spherical nematic shell with a nanoparticle represented as a green circle with the inscribed “NP”. The NP–LC interaction is characterized by μ≪1 in (**a**,**c**) and by μ≫1 in (**b**,**d**). The amplitude of nematic order $s/s_{0}$ is presented with color coding, while the orientation of molecules is denoted by the rods in the (φ,l)-plane (**a**,**b**). The degree of biaxiality $\beta^{2}$ is presented with color coding (**c**,**d**). $L_{s}$ stands for the length of the shape profile curve. The parameters used in the simulations are R/ξ=3.5, $k_{e}=k_{i}/2$, $q_{3}=0.04$.

**Figure 6 nanomaterials-14-00246-f006:**
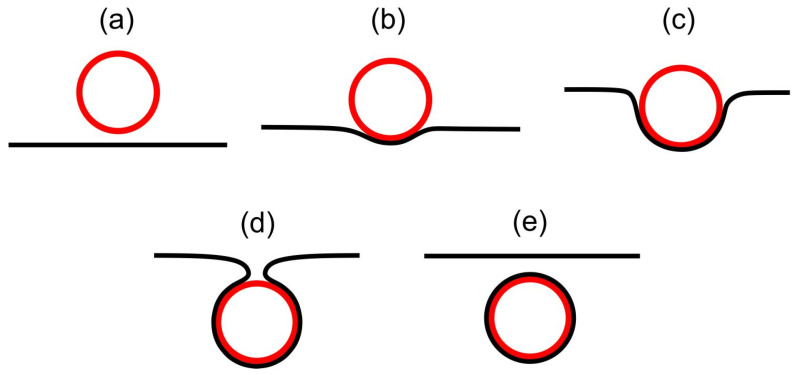
Schematic representation of the endocytosis process in biological membranes, illustrating cross-sections of the membrane around the particle in different phases of particle engulfment. Panel (**a**) shows a free particle (drawn in red) in the vicinity of the membrane (drawn in black). Panel (**b**) shows the initial stage of particle engulfment by the membrane, and panel (**c**) illustrates the progressive engulfment driven by particle-membrane adhesion/binding forces [67] and the non-homogeneous distribution of membrane constituents [3]. At the end of the engulfment process, a thin membrane neck is formed, as schematically shown in panel (**d**), which may then disappear in the process of fission [68], where the membrane-enveloped particle is detached from the membrane (see panel (**e**)).

## Data Availability

Data are contained within the article.

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
