# Peer review of "Biaxial Structures of Localized Deformations and Line-like Distortions in Effectively 2D Nematic Films"

_nanomaterials, 2024, doi:10.3390/nano14030246_

Round 1

Reviewer 1 Report

Comments and Suggestions for Authors

The authors numerically studied localized elastic distortions in effectively two-dimensional nematic shells. They used a mesoscopic Landau-de Gennes-type and limited their theoretical consideration to the axially symmetric three-dimensional shapes of nematic shells. They demonstrated that some nematic configurations are enabled by the order reconstruction mechanism.

The authors said that their results can be used for controlled assembly of NPs.

The paper can be considered as original one, well-organized, well-structured and well-written with a good cited literature.

Some point can be raised:

1-The Landau-de Gennes-type approach was mentioned in the abstract; however, we do not find any other mention in the manuscript. Authors are invited to devote a section on the above approach.

2- It seems that the results presented in this paper are very short. Could authors more develop the results by highlighting the new and original results obtained with respect to other results of literature.

3- The paper is submitted to “Nanomaterials”. I suggest to authors to devote a section to the physical applications.

The authors said in their conclusion: “For example, conductive NPs assembled within a localized line distortion could serve as a conducting wire. Furthermore, recent results reveal that line-like distortions could be rewired using external electric field. If trapped NPs follow such reconfiguration one could achieve, for example, rewritable conductive nanowires”.

This paragraph should be more developed in this manuscript with specific physical cases.

Reviewer 2 Report

Comments and Suggestions for Authors

The submitted manuscript deals with the numerical study of localized elastic distortions in effectively two-dimensional nematic shells. The presented study delves into a mesoscopic Landau-de Gennes-type approach to characterize nematic shells, focusing on the tensor nematic order parameter describing the orientational order and the curvature tensor describing the shell shape as key descriptors. The study specifically examines axially symmetric three-dimensional shapes of nematic shells. 

One notable aspect of this study is its exploration of the potential applications arising from strongly elastically deformed regions in nematic shells.  It should be pointed out that the predicted line distortion regions could act as attractors for certain types of nanoparticles.

The authors explore the potential practical applications of the observed phenomena, providing a valuable bridge between theoretical understanding and potential applications in nanotechnology. The presented discussion is supported by appropriate references and the theoretical part adequately discusses the used models. The clarity of the presentation and presented findings make it a valuable contribution to the field.

Author Response

Thank you for positive comments on our manuscript.

Reviewer 3 Report

Comments and Suggestions for Authors

This work presents a promising study on order reconstruction in nematic shells with potential for applications in NP assembly. However, the following issues should be addressed more closely. 

1. Ensure consistent terminology for distortions throughout the manuscript (e.g., "line-like distortions,").

2. Quantitative details: Consider mentioning the critical curvature required for line distortions to form, which will provide a better sense of feasibility.

3. Mechanism elaboration: Expand on the specific NP characteristics (size, surface properties) that would be suitable for DCR in the identified distortion regions.

4. Briefly discuss the potential experimental challenges and techniques for verifying the predicted NP interactions with line distortions.

Comments on the Quality of English Language

The manuscript was well written, however, please double-check it to avoid spelling mistakes. (e.g., eingevalues, line 88).

Round 2

Reviewer 1 Report

Comments and Suggestions for Authors

The author's response and the corrected version are well-done.